# Correct determination of charge transfer state energy from luminescence spectra in organic solar cells

Mathias List[1], Tanmoy Sarkar[1], Pavlo Perkhun[2], Jörg Ackermann[2], Chieh Luo[1] & Uli Würfel [1,3]

Generation and recombination of electrons and holes in organic solar cells occurs via charge transfer states located at the donor/acceptor interface. The energy of these charge transfer states is a crucial factor for the attainable open-circuit voltage and its correct determination is thus of utmost importance for a detailed understanding of such devices. This work reports on drastic changes of electroluminescence spectra of bulk heterojunction organic solar cells upon variation of the absorber layer thickness. It is shown that optical thin-film effects have a large impact on optical out-coupling of luminescence radiation for devices made from different photoactive materials, in configurations with and without indium tin oxide. A scattering matrix approach is presented which accurately reproduces the observed effects and thus delivers the radiative recombination spectra corrected for the wavelength-dependent out-coupling. This approach is proven to enable the correct determination of charge transfer state energies.

[1] Fraunhofer Institute for Solar Energy Systems (ISE), Heidenhofstr. 2, 79110 Freiburg, Germany. [2] Centre Interdisciplinaire de Nanoscience de Marseille (CINaM), Campus de Luminy, Case 913, 13288 Marseille, France. [3] Materials Research Center, University of Freiburg, Stefan-Meier-Str. 21, 79104 Freiburg, Germany. Correspondence and requests for materials should be addressed to U.W. (email: uli.wuerfel@ise.fraunhofer.de)

Organic photovoltaics (OPV) has the potential of low-cost and large-area production of flexible and light-weight solar cells. However, the power conversion efficiencies (PCE) achieved thus far are not yet competitive to commercially available technologies like, e.g., crystalline silicon solar cells. Hence, improving the PCE is a key challenge in OPV research and development. In the commonly used organic bulk hetero-junction solar cell (OSC), a charge transfer (CT) complex is formed at the donor–acceptor (D–A) interface due to interaction of the D and A molecules, resulting in an absorption and emission band at lower energies compared to the optical gaps of the individual D and A materials. Although the CT complex does not contribute significantly to the overall absorption in OSC it nevertheless is the intermediate state over which charge carrier generation and recombination occur forming a CT exciton[1–4]. The term CT state will be used here to refer to the excited state of the CT complex where the electron is located at the acceptor and the hole at the donor where the CT energy ($E_{CT}$) is the excitation energy with respect to the CT complex ground state. The energy ($E_{CT}$) and occupation of the CT states are crucial as they determine the open-circuit voltage ($V_{OC}$) and are thus directly correlated to the PCE[1,5–7], which will be discussed in more detail below.

Luminescence spectroscopy is used to investigate CT states[6,8–13]; however, the quantum yield of radiative CT recombination of typically used D–A blends in OSC lies in the range of $10^{-9}$–$10^{-6}$[6,14] and is thus not trivial to detect. Especially, electroluminescence (EL) spectroscopy is suitable to examine CT complexes since EL is dominated by CT state emission[6,15]. Therefore, this work will focus on EL, while the results are valid for any kind of radiative emission, such as photoluminescence.

For organic light-emitting diodes (OLED), microcavity effects have been reported to influence emission[16–18] and optical out-coupling is a central part of the conducted research[19,20]. Optical simulations in the field of OPV are commonly used to improve wavelength-dependent optical field distribution (incoupling of sunlight) and therefore generation[21–23]. Microcavity effects have been used to enhance quantum efficiency close to the absorption band edge and thus improve device performance[24–29] and they were furthermore utilized to alter transmission and produce color-tunable semitransparent OSCs in the visible range[30]. As a matter of fact, these OPV-related optical studies focus on the wavelength regions of the donor and acceptor absorption and thus ultimately on the optimization of the external quantum efficiency (EQE). In contrast, to the knowledge of the authors microcavity effects have not been considered yet when using luminescence spectroscopy to characterize radiative recombination occurring via CT states within OSCs, i.e., at significant longer wavelengths. For this reason, this work focuses on the wavelength region of CT state absorption and emission in OSC and takes optical out-coupling effects explicitly into account. It will be shown that it is essential to account for the optical properties of the investigated system when CT state emission is detected and analyzed by luminescence spectroscopy.

Absorption and emission spectra of CT states are stokes shifted due to the reorganization of the CT complex[31]. According to Marcus theory, the reduced CT state emission spectrum $I_f/E$ (where $I_f$ is the emission spectra and $E$ the energy) measured in photons per unit time and unit energy[1,32] is given by:

$$\frac{I_f}{E} = \frac{f_{I_f}}{\sqrt{4\pi\lambda_{RO}k_B T}}\exp\left(\frac{-(E_{CT}-\lambda_{RO}-E)^2}{4\lambda_{RO}k_B T}\right), \quad (1)$$

with $\lambda_{RO}$ being the reorganization energy, $f_{I_f}$ being a prefactor depending on the electronic coupling matrix element of the D–A system, $k_B$ the Boltzmann constant, and $T$ the temperature. $E_{CT}$

can be determined by fitting Eq. (1) to the experimentally determined EL spectra. The peak position for CT state emission is therefore expected to be $E_{CT}^{em} = \max(I_f/E) = E_{CT} - \lambda_{RO}$, i.e., red-shifted by $\lambda_{RO}$. In the same manner, CT state absorption spectra will be blue shifted leading to a characteristic Stokes shift $2\lambda_{RO}$ of ~0.4 eV[2].

Under open-circuit conditions, generation and recombination are equally large which determines the electron and hole concentrations. The open-circuit voltage of a solar cell is given by:

$$eV_{OC} = E_G - k_B T \ln\left(\frac{N_{ETL}N_{HTL}}{n_e n_h}\right) \quad (2)$$

Therein, $N_{ETL}$ and $N_{HTL}$ are the effective densities of electronic states of the electron and hole transport levels (ETL in the acceptor and HTL in the donor) and $e$ the elementary charge. For conventional semiconductors, $E_G$ resembles the band gap. In OSC, an effective gap $E_{G,eff}$ is considered as the energy difference between electrons in the ETL and holes in the HTL.

Accounting for energetic disorder and under the assumption of thermodynamic equilibrium between free charge carriers and the occupation of CT states, Burke et al. derived the relation

$$eV_{OC} = \underbrace{E_{CT} - \frac{\sigma_{CT}^2}{2k_B T}}_{E_{G,eff}} - k_B T \ln\left(\frac{ef N_0 L}{\tau_{CT}J_{rec}}\right) \quad (3)$$

and verified it to be valid for OSC[7]. For this, a Gaussian-shaped energetic CT state distribution with standard deviation $\sigma_{CT}$ and center $E_{CT}$ is assumed representing the inhomogeneously broadened CT state energies at the D–A interfaces. Here, $\tau_{CT} = 1/k_r$ is the average lifetime of the CT states, $J_{rec}$ the recombination current ($J_{rec} \approx J_{SC}$) under open-circuit conditions, $f$ is the volume fraction of the D–A interfaces, $N_0$ is the density of CT states related to the interfacial volume, and $L$ the thickness of the photoactive layer (PAL). In case of a Gaussian energetic disorder, the experimentally observed CT spectrum is shifted to $E_{CT}^{exp} = E_{CT} - \sigma_{CT}^2/2k_B T$ with $\lambda_{RO}^{exp} = \lambda_{RO} + \sigma_{CT}^2/2k_B T$. From Eq. (3), it can be seen that this experimentally observed CT state energy $E_{CT}^{exp} = E_{CT} - \sigma_{CT}^2/2k_B T$ resembles exactly the effective band gap $E_{G,eff}$ for OSC. Gaussian energetic disorder leads to a reduction of $V_{OC}$ by $\sigma_{CT}^2/2k_B T$ and thus depends on the width of the disorder distribution. Note that the energy of a particular CT complex depends on the local environment at the D–A interface, which means that not all charge carriers can thermalize down to the low energetic tail of the distribution. From Eqs. (1) and (3), it is obvious that knowledge about CT state energy $E_{CT}$ and the width of emission or absorption spectra are key for a detailed understanding and thus optimization of $V_{OC}$ in OSC. Here we show that interference effects have a significant impact on emission spectra in the relevant range and have to be taken into account when CT state energies are determined by luminescence spectroscopy.

## Results

**Optical simulation of radiative out-coupling.** Emitted electromagnetic waves are partially reflected at material interfaces in multilayer structures due to differences in their optical coefficients, i.e., refractive indices ($n$) and extinction coefficients ($k$). The direct and reflected waves superimpose leading to constructive and destructive interference. This can alter the wavelength dependency of the spectrum measured outside the device $I(\lambda)$ in comparison to the intrinsic emission line shape $I_{hom}(\lambda)$ of the emitting material, i.e., the emitted spectrum in absence of any reflecting interfaces. The radiative out-coupling factor $\gamma_{OC}$ is

defined as the quotient of these two quantities:

$$\gamma_{\mathrm{OC}}(\lambda) = \frac{I(\lambda)}{I_{\mathrm{hom}}(\lambda)} \qquad (4)$$

As the emitted radiation in the organic solar cells investigated in this study originates from the entire PAL $\bar{\gamma}_{\mathrm{OC}}$ is defined as the average out-coupling factor over all emission positions within the PAL as outlined in detail below. For simulations of the radiance enhancement within the active layer of the OSCs, an implementation[33] of the scattering matrix (S matrix) method[34,35] based on custom code was applied. Within this method, the exact configuration of the investigated OSC layer stacks can be simulated. Experimentally determined $n$ and $k$ values are used as input parameters. All used parameters can be found in the Supplementary Fig. 1, 2. Super- and substrate are air ($n = 1$) and glass ($n = 1.67$), respectively, or vice versa depending on the layer stack. When the luminescence is observed through the 1.1 mm thick glass substrates, the latter was modeled as superstrate with infinite elongation. Thereby, no reflection at the glass–air interface is included in the simulation, i.e., the back reflection at the 1.1 mm distant glass–air boundary is assumed to be not coherent and does not contribute to interference effects within the stack.

The used layer stacks are one-dimensional photonic structures, therefore a one-dimensional simulation along the direction perpendicular to the electrodes (referred to as $x$-axis) fully describes these systems. Luminescence within the photoactive layer (PAL) was implemented by a dipole-like emitter and its position $x$ was varied throughout the whole PAL. For this, the layer stack was split at position $x$ and two S matrices, one for the lower and one for the upper layers, were calculated. The out-coupled radiance $I_{\mathrm{PAL}}(x)$ from the device into the superstrate for each dipole position was obtained from the time-averaged Poynting vector $\bar{\mathbf{S}}$ with a sampling of $\Delta x = d_{\mathrm{PAL}}/N$, with $d_{\mathrm{PAL}}$ being the thickness of the photoactive layer. The error was proven to be <1% for $N \geq 100$ (relative to $N = 1000$), thus $N = 100$ was used for the results shown in this work. To obtain a spherically symmetrical emission the dipole was oriented in $x$-, $y$-, and $z$-direction successively and the resulting radiances were averaged. Under the assumption of a relative permeability of $\mu_{\mathrm{r}} = 1$ for electromagnetic plane waves $|\bar{\mathbf{S}}|$ can be calculated by the amplitude $|\mathbf{E}|$ of the electric field:[36]

$$|\bar{\mathbf{S}}| = \frac{1}{2}\sqrt{\frac{\varepsilon_0 \varepsilon_{\mathrm{r}}}{\mu_0}} |\mathbf{E}|^2 \qquad (5)$$

with $\varepsilon_0$, $\varepsilon_{\mathrm{r}}$, and $\mu_0$ being the vacuum and relative permittivity and the permeability of free space, respectively. As reference, the free space radiance $I_{\mathrm{hom}}$ of a homogeneous system with infinite elongation was calculated. To obtain the radiative out-coupling factor $\gamma_{\mathrm{OC}}(x)$ the radiance $I_{\mathrm{PAL}}(x)$ for an emission from the full stack was divided by the reference radiance $I_{\mathrm{hom}}$[37] according to Eq. (4). Assuming spatially homogeneous recombination and thus homogeneous luminescence within the PAL of an OSC, the radiative out-coupling factor of the system was calculated as spatial average of $\gamma_{\mathrm{OC}}(x)$, i.e., the spatial average over all emission positions $x$ in the PAL:

$$\bar{\gamma}_{\mathrm{OC}} = \frac{1}{d_{\mathrm{PAL}}} \int_0^{d_{\mathrm{PAL}}} \gamma_{\mathrm{OC}}(x)\, \mathrm{d}x = \frac{1}{d_{\mathrm{PAL}}} \int_0^{d_{\mathrm{PAL}}} \frac{I_{\mathrm{PAL}}(x)}{I_{\mathrm{hom}}}\, \mathrm{d}x \qquad (6)$$

Since $\gamma_{\mathrm{OC}}$ and thus $\bar{\gamma}_{\mathrm{OC}}$ are relative quantities, the amplitude of the dipole emitter can be chosen arbitrarily and was set to unity.

**Luminescence spectroscopy.** To investigate CT state emission, EL spectroscopy was performed for a variation of photoactive D–A systems and layer stacks. The EL spectrum $I_{\mathrm{EL}}(\lambda)$ measured in power per unit area and per unit wavelength for a device with a PAL consisting of P3HT:PC$_{61}$BM with PEDOT:PSS as transparent electrode (device type A in Fig. 1a is plotted as blue solid line in Fig. 2). The emission features a rather wide peak with a maximum at ~1360 nm.

After measuring the EL spectrum, the very same device (type A) was modified by depositing a 10 nm thick (and thus semitransparent) Ag layer on top of the transparent electrode and renamed to type B, depicted in Fig. 1b. When measuring EL again, drastic changes are observed in the spectrum which can be seen from the solid red line in Fig. 2. It shows a narrower peak with a central wavelength of about 1250 nm. It should be pointed

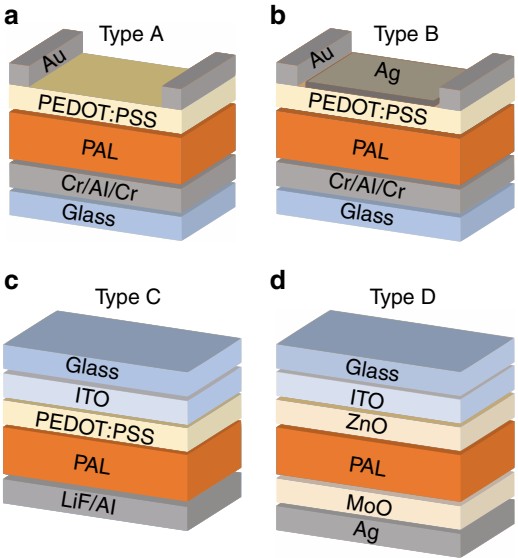

**Fig. 1** Schematic illustration of device architectures of the used organic solar cells. Layer stacks of an optically inverted OSC (**a**, **b**) with additional semitransparent Ag layer. The ITO-based architecture using PEDOT:PSS and LiF/Al electrodes is shown in **c** and inverted ITO-based architecture using ZnO and MoO$_3$ as charge selective layers in **d**

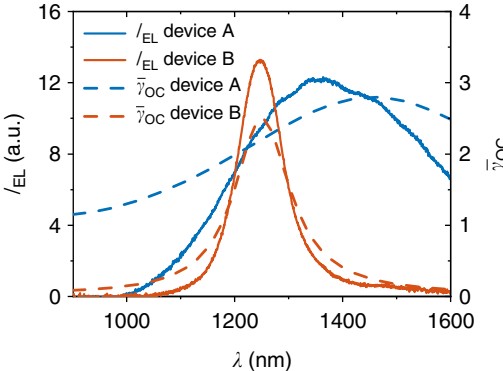

**Fig. 2** EL spectra and out-coupling factor of an OSC with and without semitransparent layer. EL spectra of a P3HT:PC$_{61}$BM cell without (type A, see Fig. 1) and with a thin, semitransparent Ag layer on top (type B) together with the calculated radiant out-coupling factors $\bar{\gamma}_{\mathrm{OC}}$. The more distinct peak with the additional semitransparent layer originates from the formation of an optical cavity what can be seen from the out-coupling factor

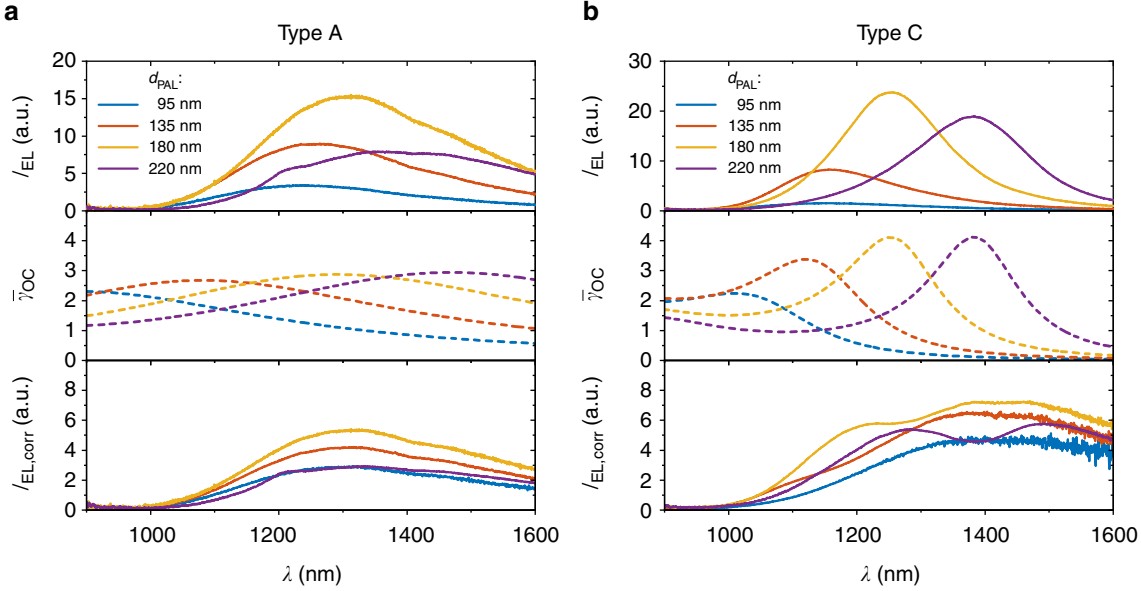

**Fig. 3** EL spectra $I_{EL}(\lambda)$ of P3HT:PC$_{61}$BM solar cells for different absorber thicknesses $d_{PAL}$. For ITO-free devices type A in **a** and ITO-based devices type C in **b**, the calculated radiant out-coupling factors $\bar{\gamma}_{OC}(\lambda)$ for the corresponding layer stacks and the spectra corrected for the optical out-coupling $I_{EL,corr} = I_{EL} \cdot \bar{\gamma}_{OC}^{-1}$

out that the PAL and contact materials were not changed and morphologic changes are not expected by mere deposition of such a thin Ag layer on top of the PEDOT:PSS layer and thus the recombination properties are also not expected to change. Accordingly, the different spectrum has to originate from the change of the optical properties of the system. In addition, the calculated radiant out-coupling factor $\bar{\gamma}_{OC}$ for the corresponding layer stacks are plotted in Fig. 2 (dashed lines). In case of device type B, the out-coupling factor $\bar{\gamma}_{OC}$ resembles the spectral shape quite accurately proving further that the spectrum is very strongly governed by out-coupling effects. Therefore, the data hardly allows for any statement about the spectral fingerprint of the emitting photoactive material, i.e., the energetics of the CT complex.

For type A, the wavelength dependence of $\bar{\gamma}_{OC}$ originates from the interference of the direct emission and the reflected portion at the back mirror, i.e., wide-angle interference. In this case, the path difference between direct and reflected light depends on the distance of the photon emission from the reflective electrode, thus, the averaging over all emission positions within the PAL leads to a quite homogeneous out-coupling. Still, aspects of the spectral shape like the asymmetry of the spectrum can be partially explained by optical out-coupling. In contrast, the additional highly reflective surface in the type B stack forms an optical microcavity leading to a pronounced peak in $\bar{\gamma}_{OC}$ (multibeam interference). While the amplitude of the out-coupled radiance $I_{PAL}(x)$ and thus $\bar{\gamma}_{OC}(x)$ still depend on the position of emission within the cavity the peak position is determined by the optical distance of the mirrors.

The previously discussed case has a more exemplary character, since a reflective surface at the illumination side reduces photon absorption and hence is avoided in the design of an OSC. The following part therefore focuses on regularly used architectures of OSC. The EL spectra of P3HT:PC$_{61}$BM devices (type A) for varying PAL thicknesses $d_{PAL}$ are shown in Fig. 3a along with the corresponding $\bar{\gamma}_{OC}(\lambda)$ and the corrected spectra $I_{EL,corr} = I_{EL} \cdot \bar{\gamma}_{OC}^{-1}$. The corrected intensity $I_{EL,corr}$ represents the expected free space radiance without the optical environment, i.e., the emission in a homogeneous medium. In Fig. 3b, the same quantities are plotted for ITO-based devices (type C in Fig. 1c).

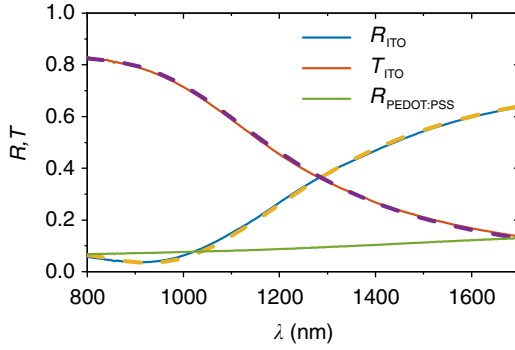

**Fig. 4** Reflectance ($R$) and transmittance ($T$) of ITO and the reflectance of PEDOT:PSS. Materials were measured on glass (solid lines) and the fits (dashed lines) to the ITO data for the determination of $n$ and $k$ values. ITO shows a strong increase in reflectance for wavelengths longer than 1000 nm

The measured spectra depend sensitively on $d_{PAL}$ and show peak positions ranging from 1240 to 1370 nm for device type A and from 1050 to 1380 nm for type C devices. In addition, peak amplitudes range from 3.4 to 15.3 for type A and from 1.5 to 23.8 for type C.

In great contrast, all-corrected intensities $I_{EL,corr}$ are quite similar regarding the shape and amplitude for the whole range of different values of $d_{PAL}$ and for the two different device architectures as could be expected for the same photoactive material used in these devices. This is a clear indication that the measured spectra are strongly influenced by the optical properties of the layer stacks. The measured spectra using PEDOT:PSS as transparent contact (type A) are wider compared to the spectra of the ITO-based devices (type C), which show more narrow peaks. For both architectures, the overall intensities vary as a function of $d_{PAL}$ but more severe for the ITO-based ones. While the out-coupling $\bar{\gamma}_{OC}$ in the PEDOT:PSS device is quite homogeneous as discussed previously, the out-coupling of the ITO-based devices feature clear peaks. These peaks originate from the high reflectance of ITO at the relevant wavelengths as shown in Fig. 4

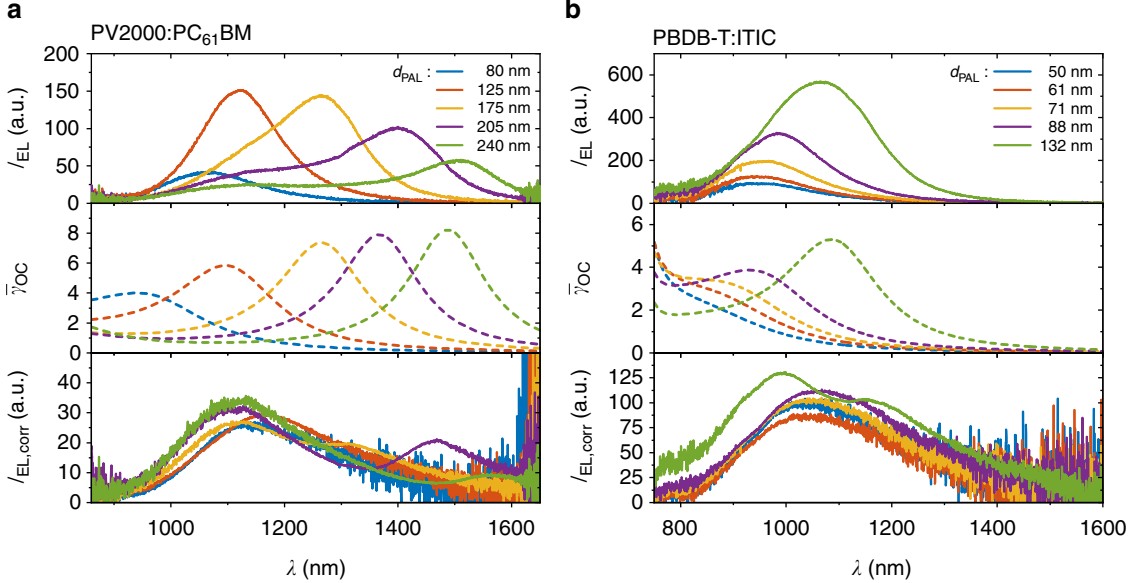

**Fig. 5** Electroluminescence spectra for PV2000:PC$_{61}$BM and PBDB-T:ITIC solar cells. EL spectra $I_{EL}(\lambda)$ of PV2000:PC$_{61}$BM solar cells (**a**) and PBDB-T:ITIC solar cells (**b**) for different thicknesses $d_{PAL}$ of the photoactive layer in an ITO-based design (type D in Fig. 1), the calculated radiant out-coupling factors $\bar{\gamma}_{OC}$ for the corresponding layer stacks and the spectra corrected for out-coupling

(measured on glass), resulting in the formation of an optical microcavity. In contrast, PEDOT:PSS (measured on glass) does not show a comparable increase of its reflectance in the infrared part of the spectrum (green curve in Fig. 4).

It should be noted that varying $d_{PAL}$ can result in morphological and also energetic changes which may influence $I_{EL}$ and thereby $I_{EL,corr}$[38]. The deviations present in the corrected spectra are due to slight offsets in peak position and peak width between measurement and simulation. The results shown in Fig. 3 clearly demonstrate the strong impact of interference effects on the observed phenomena.

Additionally, the polymer PV2000 as an example for a high-performing polymer in combination with PC$_{61}$BM and the state of the art non-fullerene DA system PBDB-T:ITIC are investigated in ITO-based solar cells[39] (full names listed under cell preparation further below). This is shown in Fig. 1d (type D) yielding PCE between 4.9% ($d_{PAL}$ = 80 nm) and 7.9% ($d_{PAL}$ = 240 nm) in the case of PV2000:PC$_{61}$BM and 7.2% ($d_{PAL}$ = 50 nm), 8.2% ($d_{PAL}$ = 132 nm), and a maximum of 9.3% ($d_{PAL}$ = 88 nm) for PBDB-T:ITIC. Electroluminescence spectra for the PV2000:PC$_{61}$BM devices are shown in Fig. 5a for layer thicknesses in the range of 80 nm ≤ $d_{PAL}$ ≤ 240 nm alongside with $\bar{\gamma}_{OC}$ and $I_{EL,corr}$. While $I_{EL}$ peaks between 1050 and 1500 nm with maximum peak intensities in the range of 40–150 (a.u.) all-corrected spectra feature a peak between 1120 and 1170 nm with maximal intensities between 26 and 34 (a.u.). In the uncorrected spectra, these devices show a shift in wavelength of the peak of almost 500 nm and a significant change in overall emission for different thicknesses of the absorber layer. This can clearly be ascribed as to originate from changes in the optical out-coupling properties upon variation of absorber thickness. The spectra $I_{EL}$, $I_{EL,corr}$, and $\bar{\gamma}_{OC}$ for the PBDB-T:ITIC devices for absorber thicknesses 50 nm ≤ $d_{PAL}$ ≤ 132 nm are shown in Fig. 5b. Thicker photoactive layers could not be realized due to a poor resulting film quality. $I_{EL}$ peaks between 940 and 1070 nm with maximum peak intensities in the range of 100–600 (a.u.), whereas all-corrected spectra $I_{EL} \cdot \bar{\gamma}_{OC}^{-1}$ feature a peak at ~1070 nm with maximal intensities between 85 and 12 (a.u.). Note that the strong s-shape in the corrected spectrum of the device with an average absorber thickness of 132 nm originates from strong fluctuations

of $d_{PAL}$ of about 30 nm (as measured by profilometry), resulting in an inaccurate correction when only the average thickness is taken into account.

OSCs usually feature a reflective back contact. In case of no interference in the device, i.e., incoherent emission of light, $\gamma_{OC}$ equals 2 since the radiation is reflected at the backside mirror and emitted only into one hemisphere. Hence, the total emitted power which is proportional to the square of the electric field of the wave $|\mathbf{E}|^2$ remains unchanged. In contrast, for coherent light, $|\mathbf{E}|$ is doubled when the criterion for constructive interference is fulfilled. For this reason, the radiance is enhanced by a factor of 4 compared to the free space emission thus resulting in a factor of 2 in total emitted power. As could be shown experimentally by Drexhage, this doubling of the total emitted power is accompanied by a decreased radiative lifetime for the emission[40]. This decreased radiative lifetime for emission does also apply for the devices investigated in this study. However, the total recombination and thus the lifetime of electrons and holes in typical organic solar cells are strongly dominated by non-radiative recombination, which can clearly be seen from the previously mentioned low-quantum yields for radiative CT recombination in the range of only $10^{-9}$–$10^{-6}$ in typical D–A systems. For this reason, interference effects do not have any measurable impact on the overall charge carrier lifetimes in these materials.

For the determination of the CT state energy, the previously shown spectra $I_{EL}(\lambda)$ (in W/nm/m$^2$) were converted to $\tilde{I}_f(E)$ given in photons per unit spectral energy and per area by dividing by the photon energy $E^3$[41]. The normalized reduced spectra $I_f/E$ are plotted in Fig. 6 alongside with the corrected spectra $I_{f,corr}/E$ for all systems. The corrected spectra of PBDB-T:ITIC are smoothened by a floating average over 20 data points to reduce noise and to ensure proper normalization. Gaussian functions are fitted by means of Eq. (1) to all spectra and added to Fig. 6. The resulting CT state energy $E_{CT}$, reorganization energy $\lambda_{RO}$ and the energy of maximum CT emission $E_{CT}^{em} = \max(I_f/E)$ from the fits for all systems and the corresponding photoactive layer thicknesses are listed in Table 1.

$E_{CT}^{em}$ is strongly influenced by interference as previously seen from the wavelength-dependent spectra in Figs. 3 and 5. From Fig. 6 it gets obvious that the high energetic tails, i.e., the apparent

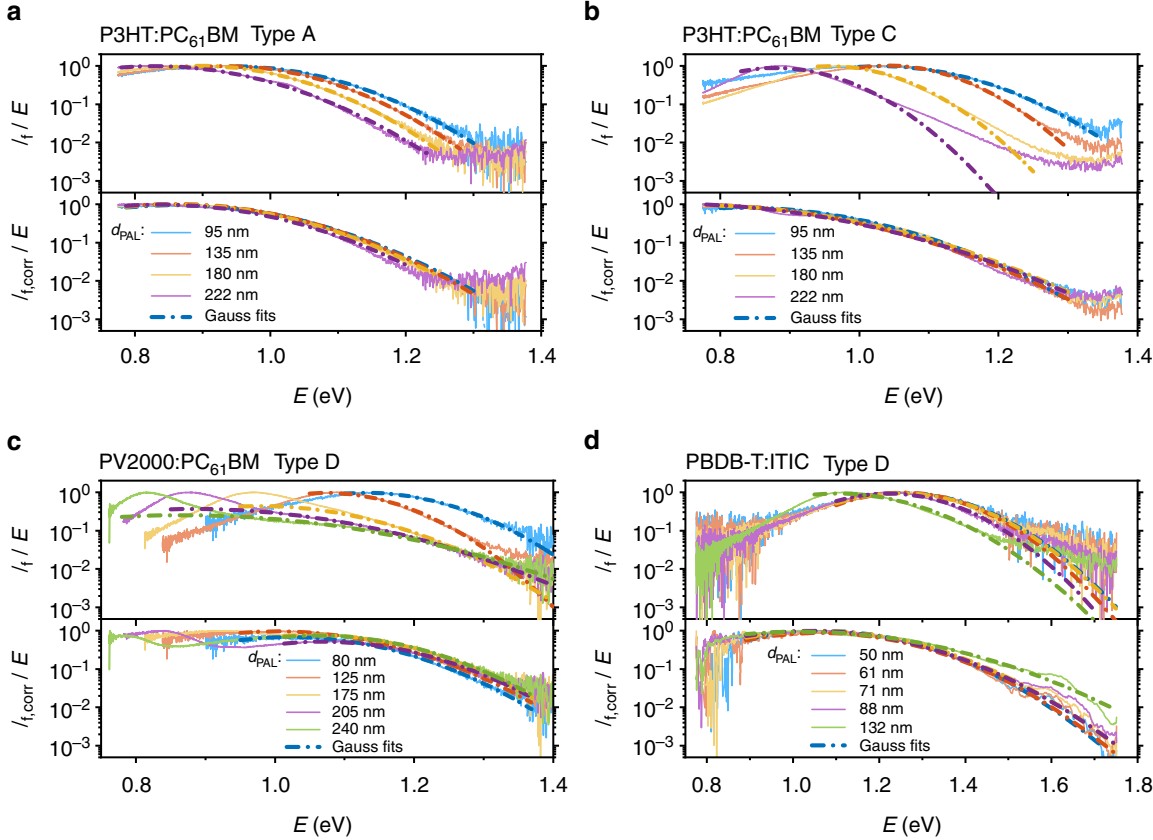

**Fig. 6** Reduced emission spectra $I_f/E$ and corrected spectra $I_{f,corr}/E$ data (solid lines). Gaussian fits (dashed lines) given by Eq. (1) for P3HT:PC$_{61}$BM devices of type A in **a** and P3HT:PC$_{61}$BM devices of type C in **b** as well as type D devices made from PV2000:PC$_{61}$BM (**c**) and PBDB-T:ITIC (**d**)

**Table 1 Results of Gaussian fits to the experimental and corrected reduced EL spectra for different photoactive materials and varying absorber thickness $d_{PAL}$ plotted in Fig. 6**

| $d_{PAL}$ (nm) | $E_{CT}^{em}$ (eV) | $\lambda_{RO}$ (eV) | $E_{CT}$ (eV) | $E_{CT,corr}^{em}$ (eV) | $\lambda_{RO,corr}$ (eV) | $E_{CT,corr}$ (eV) |
|---|---|---|---|---|---|---|
| P3HT:PC$_{61}$BM (type A) | | | | | | |
| 95 | 0.947 | 0.262 | 1.209 | 0.860 | 0.364 | 1.224 |
| 135 | 0.934 | 0.233 | 1.167 | 0.858 | 0.350 | 1.208 |
| 180 | 0.896 | 0.236 | 1.133 | 0.856 | 0.345 | 1.201 |
| 220 | 0.828 | 0.300 | 1.128 | 0.837 | 0.353 | 1.190 |
| P3HT:PC$_{61}$BM (type C) | | | | | | |
| 95 | 1.028 | 0.227 | 1.256 | 0.830 | 0.388 | 1.218 |
| 135 | 1.040 | 0.131 | 1.171 | 0.764 | 0.493 | 1.256 |
| 180 | 0.957 | 0.131 | 1.089 | 0.762 | 0.540 | 1.302 |
| 220 | 0.880 | 0.127 | 1.007 | 0.727 | 0.594 | 1.321 |
| PV2000:PC$_{61}$BM (type D) | | | | | | |
| 80 | 1.130 | 0.190 | 1.320 | 1.021 | 0.270 | 1.292 |
| 125 | 1.072 | 0.152 | 1.224 | 1.001 | 0.307 | 1.308 |
| 175 | 0.977 | 0.301 | 1.278 | 1.051 | 0.262 | 1.313 |
| 205 | 0.881 | 0.565 | 1.446 | 1.067 | 0.263 | 1.330 |
| 240 | 0.868 | 0.718 | 1.586 | 1.060 | 0.259 | 1.319 |
| PBTB-T:ITIC (type D) | | | | | | |
| 50 | 1.251 | 0.353 | 1.604 | 1.067 | 0.586 | 1.653 |
| 61 | 1.254 | 0.308 | 1.562 | 1.062 | 0.620 | 1.682 |
| 71 | 1.249 | 0.351 | 1.600 | 1.056 | 0.666 | 1.723 |
| 88 | 1.222 | 0.316 | 1.538 | 1.039 | 0.696 | 1.736 |
| 132 | 1.101 | 0.455 | 1.556 | 0.997 | 1.168 | 2.166[a] |

[a]Value not reliable due to layer thickness fluctuation (see text and Fig. 5b)

reorganization energy $\lambda_{RO}$, and thus the estimated CT energy $E_{CT} = E_{CT}^{em} + \lambda_{RO}$ are significantly affected by interference effects as well. The spectra of the ITO-based P3HT:PC$_{61}$BM and PBDB-T:ITIC devices (Fig. 6b–d) show that the uncorrected data sets differ from the corrected spectra for all absorber thicknesses and in case of P3HT:PC$_{61}$BM for $d_{PAL} > 135$ nm microcavity effects can result in observed spectra significantly deviant from the expected Gaussian shape. This results in a strong dependence of the fit results on the chosen data range making a fit of a Gaussian function questionable in general. In case of PV2000:PC$_{61}$BM (Fig. 6c), the uncorrected spectra look like a superposition of two peaks for $d_{PAL} \geq 175$ nm due to the resonant peak of the microcavity. These strongly altered spectral shapes make a meaningful analysis of the uncorrected data difficult and prone to misinterpretation. Interestingly, there are cases where two effects compensate each other when the CT state energy $E_{CT} = E_{CT}^{em} + \lambda_{RO}$ is determined by the applied fitting procedure. As an example, the uncorrected PBDB-T:ITIC spectra feature higher $E_{CT}^{em}$ values with lower $\lambda_{RO}$ (peak width) compared to the corrected spectra. This results in comparable CT energies although the values for $E_{CT}^{em}$ and $\lambda_{RO}$ differ significantly. Hence, even in this case the correction procedure is indispensable to achieve a deeper understanding of the underlying processes and material properties.

It can be seen in Table 1 that the corrected spectra resemble each other quite accurately for all absorber thicknesses and the fitted $E_{CT,corr}$ coincide within <0.1 eV. In contrast, the raw $E_{CT}$ energies deviate up to more than 0.35 eV for the investigated systems (note that it could be even higher for other systems). This deviation should by no means be misinterpreted as an upper limit for the error of the CT state energy as quantities derived from

uncorrected spectra are not meaningful if interference effects are dominant. In general, cell architectures featuring ITO as transparent electrode show more pronounced microcavity effects due to the high reflectivity of ITO for wavelengths longer than 1000 nm as explained in detail before. The described effects are expected to be negligible if devices which do not differ significantly in terms of their optical properties are compared among each other as could be the case when different interlayers with comparable optical constants are compared or when mere morphological changes are induced by additives or annealing procedures. Nevertheless, consistent layer thicknesses should be ensured in such cases. In contrast, correction for optical effects is crucial if the actual spectral shape of the luminescence spectrum is of interest in organic thin-film devices.

It should be stated here, since out-coupling of light is ultimately the reverse process of light absorption, the results will be affected in a comparable manner if low energetic CT state absorption tails are analyzed.

## Discussion

CT states are crucial for the thorough analysis of organic solar cells as their energy and occupation determine the open-circuit voltage ($V_{OC}$) and thus have a large impact on the achievable power conversion efficiency. For this reason, an exact determination of the CT state energy is of utmost importance. CT state energies are experimentally determined by different methods which however are always based—directly or indirectly—on absorption and emission of light. That the CT state emission can be largely influenced by the optical properties of the device in the corresponding wavelength region is shown in this work for a variety of different bulk heterojunction organic solar cells based on four different device architectures, three different photoactive materials, and various thicknesses of the absorber layer. For each device, an electroluminescence spectrum was recorded and it was found that the spectra for a given photoactive material differ strongly for different device architectures and different thicknesses of the absorber layer. In harsh contrast, the CT state energy as a fundamental property of the donor/acceptor blend is expected to be rather independent of the exact device configuration and the thickness of the absorber layer. Therefore, the optical out-coupling properties of the different devices were calculated by means of a scattering matrix formalism based on experimentally determined optical diffraction and extinction coefficients. After correction of the luminescence spectra for the specific out-coupling properties, all spectra for each photoactive material were very similar. These findings demonstrate clearly that the measured spectra are strongly influenced by microcavity effects and that the described correction for the optical out-coupling is indispensable in order to achieve reliable information about the nature of the emissive states within the photoactive layer. It is important to note that the described correction procedure is required for the correct analysis of CT state absorption spectra.

## Methods

**Optical constants**. Reflection and transmission (RT) spectra were measured for the utilized materials using a PerkinElmer LAMBDA 950 UV/Vis Spectrophotometer equipped with an integrating sphere. The corresponding layer thicknesses were measured with a Veeco Dektak 150 surface profilometer. The refractive indices ($n$) and extinction coefficients ($k$) were determined by fitting the measured RT spectra using the software SCOUT from W. Theiss Hard- and Software. The values for Ag[42], MoO$_3$[43], and ZnO[44] were obtained from literature. All $n$ and $k$ values used within this study are plotted in the Supplementary Figures 1, 2.

**Cell preparation**. Three different blend solutions for the photoactive layer were used. (i) Poly(3-hexylthiophen-2,5-diyl):[6,6]-phenyl-C61-butyric acid methyl ester (P3HT:PC$_{61}$BM) (25:17.5 mg/ml in o-dichlorobenzene), (ii) PV2000:PC$_{61}$BM (12:18 mg/ml in o-xylene), and (iii) Poly[(2,6-(4,8-bis(5-(2-ethylhexyl)thiophen-2-yl)-benzo[1,2-b:4,5-b′]dithiophene))-alt-(5,5-(1′,3′-di-2-thienyl-5′,7′-bis(2-ethylhexyl)benzo[1′,2′-c:4′,5′-c′]dithiophene-4,8-dione))]:3,9-bis(2-methylene-(3-(1,1-dicyanomethylene)-indanone))-5,5,11,11-tetrakis(4-hexylphenyl)-dithieno[2,3-d:2′,3′-d′]-s-indaceno[1,2-b:5,6-b′]dithiophene)) (PBDB-T:ITIC) (10:10 mg/ml in 97% vol. chlorobenzene and 3% vol. 1-chloronaphthalene). The polymeric donor materials were purchased from (i) Riecke; (ii) Raynergy Tek; and (iii) Solarmer Energy. The fullerene derivatives were purchased from Solenne and the non-fullerene acceptor ITIC was purchased from Solarmer Energy. The materials used for the charge selective interlayers were 2.5% wt. zinc oxide (ZnO) nanoparticle solution (Avantama N-11) purchased from Avantama AG, diluted to the final concentration of 1% wt. in 2-propanol (99.5 anhydrous), and the poly(3,4-ethylenedioxythiophene) polystyrene sulfonate (PEDOT:PSS) formulations Clevios™ AI 4083 and Clevios™ FHC Solar purchased from Heraeus. The layer stack and the thicknesses of the different layers for the ITO-based P3HT:PC$_{61}$BM cells were glass/ITO (230 nm)/PEDOT:PSS (Clevios™ AI 4083, 30 nm)/P3HT:PC$_{61}$BM/LiF (0.3 nm)/Al (100 nm). In case of PV2000, the used architecture is glass/ITO (180 nm)/ZnO (25 nm)/PV2000:PC$_{61}$BM/MoO$_3$ (10 nm)/Ag (100 nm). As for PBDB-T, the used architecture is glass/ITO/ZnO (20 nm)/PBDB-T:ITIC/MoOx (2 nm)/Ag (100 nm). In these architectures, PEDOT:PSS solutions were spun cast at 3000 rpm and annealed at 130 °C for 15 min under nitrogen, ZnO diluted solutions were spun cast at 4000 rpm and annealed at 120 °C for 10 min under nitrogen. Different layer thicknesses were achieved by a variation in spin-coating speed. The photoactive layers were annealed under nitrogen at 130 °C for 15 min (P3HT:PC$_{61}$BM), at 80 °C for 2 min, 130 °C for 5 min, and 110 °C for 2 min (PV2000:PC$_{61}$BM), and 140 °C for 15 min (PBDB-T:ITIC). The ITO-free layer stack for P3HT:PC$_{61}$BM was glass/Cr (5 nm)/Al (100 nm)/Cr (5 nm)/P3HT:PC$_{61}$BM/PEDOT:PSS (Clevios™ FHC Solar, 70 nm)/Au (grid, 100 nm). For these samples, PEDOT:PSS solution was spun cast at 3000 rpm and annealed together with the photoactive layer at 150 °C for 10 min under nitrogen. LiF, MoO$_3$, Ag, Au, and Al were all thermally evaporated at a pressure below $5 \times 10^{-6}$ mbar.

**Luminescence spectroscopy**. The setup consists of a Czerny-Turner spectrograph (Andor Shamrock 193i) with an InGaAs photo diode array (Andor iDus DU490A), which was cooled to −70 °C. Using a collimating system consisting of two lenses with an angle of acceptance of ~10° (full angle), the samples were imaged to the plane of the entrance slit of the spectrograph. A 850-nm-long pass color glass filter was used to exclude the presence of peaks originating from the second order of diffraction from the grating. All measured spectra were corrected for the spectral sensitivity of the setup using a tungsten halogen lamp as a reference standard (reference spectrum was measured by Fraunhofer ISE CalLab). Although the absolute intensities are unknown all measured spectra $I_{EL}(\lambda)$ have the same relative unit proportional to W/m$^2$/nm and can be compared among each other. The working point of the OSC was controlled by a Keithley 2400 sourcemeter. EL spectra were measured at an applied forward current density of 75 mA/cm$^2$ (P3HT:PC$_{61}$BM), 108.1 mA/cm$^2$ (PV2000:PC$_{61}$BM), and 100 mA/cm$^2$ (PBDB-T:ITIC) while the samples were produced, stored, and measured in an inert nitrogen atmosphere. The width of the entrance slit of the spectrograph was 2 mm for P3HT:PC$_{61}$BM and 0.5 mm for PV2000:PC$_{61}$BM and PBDB-T:ITIC and the height of the detector array was 0.2 mm.

**Code availability**. The computer code used during the current study is not publicly available.

## Data availability

The data that support the findings of this study are available from the corresponding author upon request.

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

## Acknowledgements

M.L. acknowledges the support from the Dr. Ruth Heerdt-Stiftung. The authors thank Florina Bödicker for the implementation of the S-matrix code and Oliver Höhn for guidance during the optical simulations. P.P. and J.A. received funding from the European Union's Horizon 2020 research and innovation program under the Marie Skłodowska-Curie Grant Agreement No. 713750. They further acknowledge support of the Regional Council of Provence-Alpes-Côte d'Azur, A*MIDEX (No. ANR-11-IDEX-0001-02), and the Investissements d'Avenir project funded by the French Government, managed by the French National Research Agency (ANR). U.W. acknowledges financial support by the German Federal Ministry for Economic Affairs and Energy under contract number 0324214G.

## Author contributions

The research idea and design of experiments for its confirmation were made by M.L. and U.W. The optimization of the PBDB-T:ITIC device configuration was carried out by P.P. under the supervision of J.A. Most solar cells used for this study were built by C.L. Electroluminescence experiments were performed by M.L. and T.S., the optical simulations were carried out by M.L. The data were analyzed by M.L. and the manuscript was written by M.L. and U.W.

## Additional information

**Competing interests:** The authors declare no competing interests.

