## [Peer Review File · Nature Communications]

Reviewers' comments:

Reviewer #1 (Remarks to the Author):

This is a clearly written, easy to follow paper which describes the procedure to correctly determine the luminescence spectrum of organic photodiodes. Since these devices have thicknesses smaller than or on the order of the wavelength of the emitted light, interference effects play a large role. Authors show nicely the optical cavity effects, and the luminescence peak shifts accompanying these. The correction procedure is straightforward, but requires knowledge of the optical constants of all materials present in the device stack.

Electroluminescence measurements of organic photovoltaic devices are becoming more popular to investigate charge transfer states, of which the properties are important for the device operation. Interference effects have not been taken into account in these types of characterization. Therefore this excellent paper is very important for OPV experts and proper device characterization, but might be of less interest for a broad readership.

Authors state in the abstract that a correction for the wavelength dependent outcoupling is indispensable for a correct determination of the charge transfer state emission energy. Quantitatively the importance of the correction is however not fully clear: Information about CT state properties is often extracted from the CT emission tails and/or CT absorption tails, requiring logarithmic plots of the emission spectra. It is unclear how much the high energy emission tails, and thus the determination of the CT state energy, are affected by these interference effects. Also, is CT absorption equally affected as emission? One of the advantages of electroluminescence measurements is that they are relatively fast. Corrections require the knowledge of n and k values of the active layer, which might be time consuming. Therefore, a simplified correction procedure, or guidelines on when corrections are absolutely necessary for the estimation of CT state properties would be very useful. For example, it would be useful to know for commonly used electrodes in which wavelength range are the largest corrections expected.

Minor: In the introduction, it is written that "...electrons and holes form a charge transfer complex by Coulombic interaction...resulting in an absorption and emission band...". Possibly it's a matter of definition, but as far as I can see, Coulomb interaction is not necessary to produce absorption and emission bands, and is thus not the key feature of a CT complex. There are definitions of CT state (<https://goldbook.iupac.org/html/C/C01006.html>) and CT complex (<https://goldbook.iupac.org/html/C/C01003.html>), by IUPAC, but none of them involves Coulomb interaction. The CT complex is the overall donor-acceptor complex (with neutral ground state), while the CT state is the excited CT complex, i.e. the state for which the charge is transferred with respect to the ground state.

Reviewer #2 (Remarks to the Author):

List et al. used optical simulations to show that the electroluminescence (EL) spectrum of an organic solar cell (OSC) depend strongly on the device structure (4 types, ITO-based and ITO-free). As the

authors explained, this finding is significant for organic solar cell research because EL spectroscopy is often employed to probe the energy of charge transfer (CT) states, which is crucial to the fundamental operation of OSC. Without accounting for such optical effects, one may obtain a CT energy that is far from accurate. In this study, the authors studied how EL spectrum of two common OSC systems, P3HT:PCBM and PV2000:PCBM, is affected by the choice of electrodes because of the micro-cavity effect. However, while the effect of micro-cavity on EL spectrum is not usually considered in OSC research, it is widely known to be important for OLED research. As the authors pointed out, there have already been many research on optimizing the emission outcoupling of OLED via optical simulation studies. Therefore, I consider this study more suitable for a specialist journal.

Here are some additional comments:

1. I am confused with the reference of PV2000 as the state-of-the-art polymer despite its 5.5% efficiency achieved when blended with PCBM. There are numerous examples of more efficient materials than PV2000. What is the chemical structure of PV2000?
2. More efficient OSC materials tend to have higher CT energy with emission closer to the optical bandgap (~800-1000nm). It would be useful to comment on the outcoupling factor at this spectral region (and its dependence on film thickness) since it will be more relevant to efficient material systems.

Reviewer #3 (Remarks to the Author):

In this manuscript, List et al. report on strong changes in the CT state emission spectrum of OSCs under different device architectures. They show that these changes are primarily due to changes in the optical properties of these devices rather than any changes in the emitting material blend itself. The authors present a scattering matrix approach that allows to reconstruct the intrinsic emission spectrum of the CT state and show that all different device architectures indeed have a similar CT state spectrum.

The paper is well written and I agree with most of the conclusions of the manuscript. However, I do question the novelty and the impact of the findings of this work. Light trapping and optical engineering in OSCs has been investigated thoroughly and the outcoupling of luminescence is ultimately the inverse process of sunlight incoupling into the device (DOIs: 10.1016/j.mattod.2014.05.008, 10.1063/1.4807000, 10.1063/1.2789677 as some examples). Furthermore, the conclusion section seems rather poor. The authors discussed the impact of the CT state energy on the open-circuit voltage in the introduction but do not comment on this anymore in the discussion. What is the impact of such broad CT state spectra on the V_{OC} ?

Further points:

- 1) The authors mention that such optical effects will also affect transient experiments, i.e. the lifetime of the state. Is there a way for the authors to demonstrate this experimentally?

2) In samples C and D, the glass substrate is also part of the layer stack (whereas in samples A and B the metal electrode does not allow light to enter the glass). Is the interference in the glass substrate considered in the scattering matrix model?

3) The definitions of I and I_{hom} (and therefore of γ_{OC}) are not particularly precise. The authors have to revise this. I could be named “the intensity spectrum measured outside the device” whilst I_{hom} might be named “intrinsic emission line shape”

4) Page 2, figure 2 and lines 6 – 7: The authors have to state here how γ_{OC} is calculated (i.e. refer to the methods section). Maybe, the authors even want to consider moving the optical simulation section into the results text since most of their conclusion rely on it.

5) Figure 4: It is stated in the text that this data was measured on films on glass. The authors should also state at which interface these reflectivities were measured. If it was measured on the electrode-air interface: Was the data corrected for the reflection on the electrode-glass interface?

6) Can the authors give a few more details on their model to make their results more reproducible? I think that the modelling section is a central part of this manuscript. Also, the authors might want to consider providing all input parameters such as the refractive indices and the layer thicknesses of all layers in the supplementary material.

Reviewers' comments:

Reviewer #1 (Remarks to the Author):

This is a clearly written, easy to follow paper which describes the procedure to correctly determine the luminescence spectrum of organic photodiodes. Since these devices have thicknesses smaller than or on the order of the wavelength of the emitted light, interference effects play a large role. Authors show nicely the optical cavity effects, and the luminescence peak shifts accompanying these. The correction procedure is straightforward, but requires knowledge of the optical constants of all materials present in the device stack.

Electroluminescence measurements of organic photovoltaic devices are becoming more popular to investigate charge transfer states, of which the properties are important for the device operation. Interference effects have not been taken into account in these types of characterization. Therefore this excellent paper is very important for OPV experts and proper device characterization, but might be of less interest for a broad readership.

Authors state in the abstract that a correction for the wavelength dependent outcoupling is indispensable for a correct determination of the charge transfer state emission energy.

Quantitatively the importance of the correction is however not fully clear: Information about CT state properties is often extracted from the CT emission tails and/or CT absorption tails, requiring logarithmic plots of the emission spectra. It is unclear how much the high energy emission tails, and thus the determination of the CT state energy, are affected by these interference effects. Also, is CT absorption equally affected as emission? One of the advantages of electroluminescence measurements is that they are relatively fast. Corrections require the knowledge of n and k values of the active layer, which might be time consuming. Therefore, a simplified correction procedure, or guidelines on when corrections are absolutely necessary for the estimation of CT state properties would be very useful. For

example, it would be useful to know for commonly used electrodes in which wavelength range are the largest corrections expected.

Minor: In the introduction, it is written that "...electrons and holes form a charge transfer complex by Coulombic interaction...resulting in an absorption and emission band...". Possibly it's a matter of definition, but as far as I can see, Coulomb interaction is not necessary to produce absorption and emission bands, and is thus not the key feature of a CT complex. There are definitions of CT state (<https://goldbook.iupac.org/html/C/C01006.html>) and CT complex (<https://goldbook.iupac.org/html/C/C01003.html>), by IUPAC, but none of them involves Coulomb

interaction. The CT complex is the overall donor-acceptor complex (with neutral ground state), while the CT state is the excited CT complex, i.e. the state for which the charge is transferred with respect to the ground state.

Answer:

We are glad to read that the reviewer considers our work as excellent. The reviewer asks about the impact of the correction procedure on the CT state energy when high energy tails are fitted (in a logarithmic plot). We performed such an analysis and yes, it has a clear impact on the resulting CT state energy, as can be seen from the (new) Figure 6. And yes, CT absorption is expected to be affected in the same manner. Note that we can say this of course only for the wavelength range for which the outcoupling was determined. As in the procedure mentioned by the reviewer the crossing of the low energy tail of the absorption with the high energy tail of the emission is determined we can safely state that CT absorption in this wavelength regime will also be affected in a similar way as found in our emission spectra. We have included this discussion in the revised version of the manuscript. We also amended a guideline about when the effects of outcoupling are expected to be most severe. Note that to the best of our knowledge the determination of the n & k values is indispensable if spectra need to be corrected as otherwise the optical properties of the system cannot be determined quantitatively. However, in our lab this is not very time consuming. Finally, we corrected the use of the terms "CT complex" and "CT state" according to the advice of the reviewer.

Reviewer #2 (Remarks to the Author):

List et al. used optical simulations to show that the electroluminescence (EL) spectrum of an organic solar cell (OSC) depend strongly on the device structure (4 types, ITO-based and ITO-free). As the authors explained, this finding is significant for organic solar cell research because EL spectroscopy is often employed to probe the energy of charge transfer (CT) states, which is crucial to the fundamental operation of OSC. Without accounting for such optical effects, one may obtain a CT energy that is far from accurate. In this study, the authors studied how EL spectrum of two common OSC systems, P3HT:PCBM and PV2000:PCBM, is affected by the choice of electrodes because of the micro-cavity effect. However, while the effect of micro-cavity on EL spectrum is not usually considered in OSC research, it is widely known to be important for OLED research. As the authors pointed out, there have already been many research on optimizing the emission outcoupling of OLED via optical simulation studies. Therefore, I consider this study more suitable for a specialist journal.

Here are some additional comments:

1. I am confused with the reference of PV2000 as the state-of-the-art polymer despite its 5.5% efficiency achieved when blended with PCBM. There are numerous examples of more efficient materials than PV2000. What is the chemical structure of PV2000?
2. More efficient OSC materials tend to have higher CT energy with emission closer to the optical bandgap (~800-1000nm). It would be useful to comment on the outcoupling factor at this spectral region (and its dependence on film thickness) since it will be more relevant to efficient material systems.

Answer:

1. The reviewer is right. We have removed the expression “state-of-the-art” polymer, have rebuilt and optimized the devices with PV2000, now yielding power conversion efficiencies up to 7.9%. Unfortunately, the supplier Raynergy Tek from Taiwan does not disclose the chemical structure of PV2000.

2. Further we have included an additional material system, a high performance polymer blended with a non-fullerene acceptor (PBDB-T:ITIC), yielding more than 9% power conversion efficiency. This absorber emits at shorter wavelength, closer to the optical bandgap. Also for this new material system we find that the spectra have to be corrected according to our proposed procedure to obtain reliable values for the CT state energy.

Reviewer #3 (Remarks to the Author):

In this manuscript, List et al. report on strong changes in the CT state emission spectrum of OSCs under different device architectures. They show that these changes are primarily due to changes in the optical properties of these devices rather than any changes in the emitting material blend itself. The authors present a scattering matrix approach that allows to reconstruct the intrinsic emission spectrum of the CT state and show that all different device architectures indeed have a similar CT state spectrum.

The paper is well written and I agree with most of the conclusions of the manuscript. However, I do question the novelty and the impact of the findings of this work. Light trapping and optical engineering in OSCs has been investigated thoroughly and the outcoupling of luminescence is ultimately the inverse process of sunlight incoupling into the device (DOIs: 10.1016/j.mattod.2014.05.008, 10.1063/1.4807000, 10.1063/1.2789677 as some examples). Furthermore, the conclusion section seems rather poor. The authors discussed the impact of the CT state energy on the open-circuit voltage in the introduction but do not comment on this anymore in the discussion. What is the impact of such broad CT state spectra on the V_{OC} ?

Further points:

1) The authors mention that such optical effects will also affect transient experiments, i.e. the lifetime of the state. Is there a way for the authors to demonstrate this experimentally?

2) In samples C and D, the glass substrate is also part of the layer stack (whereas in samples A and B the metal electrode does not allow light to enter the glass). Is the interference in the glass substrate considered in the scattering matrix model?

3) The definitions of I and I_{hom} (and therefore of γ_{OC}) are not particularly precise. The authors have to revise this. I could be named “the intensity spectrum measured outside the device” whilst I_{hom} might be named “intrinsic emission line shape”

4) Page 2, figure 2 and lines 6 – 7: The authors have to state here how γ_{OC} is calculated (i.e. refer to the methods section). Maybe, the authors even want to consider moving the optical simulation section into the results text since most of their conclusion rely on it.

5) Figure 4: It is stated in the text that this data was measured on films on glass. The authors should also state at which interface these reflectivities were measured. If it was measured on the electrode-air interface: Was the data corrected for the reflection on the electrode-glass interface?

6) Can the authors give a few more details on their model to make their results more reproducible? I think that the modelling section is a central part of this manuscript. Also, the authors might want to consider providing all input parameters such as the refractive indices and the layer thicknesses of all layers in the supplementary material.

Answer:

The reviewer criticizes a lack of novelty and gives examples in literature where optical incoupling was considered in detail in order to optimize the generation and thus the short circuit current. We agree with the reviewer that outcoupling is the reverse process of sunlight incoupling. However, sunlight incoupling in OSC matters in a wavelength region where the donor and/or acceptor materials show rather strong absorption, i.e., where by far most of the charge carrier generation occurs. This wavelength region is typically at (much) shorter wavelengths than CT absorption and especially emission. Therefore the publications mentioned by the reviewer focus on the optimization of the EQE and do not include the wavelength range in which the optical outcoupling properties are analyzed in our work. Therefore we think that our work is indeed novel as the mentioned studies address a different phenomenon. We intend to show how important it is to perform the correction of the EL spectra in order to arrive at correct values for the CT state energy and not how absorption in the shorter wavelength region can be optimized. We further show in our work why this correct determination is of utmost importance to deepen the understanding about fundamental energetic properties which have a large impact on the obtainable open-circuit voltage and thus on the power conversion efficiency.

We agree with the reviewer that the conclusion section was rather poor. We have extensively revised it and we also added a new section "CT state properties and photovoltage" in the revised manuscript about the relationship between CT state energy and open-circuit voltage also considering the effect of broad CT state spectra.

1) Although the radiative lifetime is indeed expected to be affected the overall (and thus measurable) lifetime will hardly change as it is most dominantly determined by non-radiative recombination. We have amended a corresponding text in the manuscript.

2) No, we have also added this to the text.

3) The definitions were changed according to the suggestion of the reviewer.

4) We have moved the section about optical simulation to the results section.

5) No, the data was not corrected and we have mentioned this now in the manuscript.

6) We have added more explanation on the optical model. Note that all layer thicknesses were already given in the manuscript. We have now included all n&k-values in the supplementary material.

REVIEWERS' COMMENTS:

Reviewer #1 (Remarks to the Author):

Authors have revised their paper substantially based on the referee comments. They have extended their study to real state of the art devices and they have clarified why it is important to include microcavity effects: Even though these effects are widely known in OLED research, they have been neglected for the extraction of CT state properties in OPV. Therefore, I feel the work is sufficiently important for publication in nature communications. In the revised version the impact on the determination of the CT state energy is now much more clear, although a bit hidden in the supporting information. It would be nice if the authors could put a number (0.1 eV but not 0.5 eV ?, and there is still a scattering of 50 to 100 meV on the CT state energy of corrected spectra) on the error made in the determination of the CT state energy when neglecting cavity effects, and mention these numbers in the main text. The errors seems to be smaller when thin films are used.

Minor:

- In the paragraphs on “CT state properties and photovoltage”, above equation (3), the last name of the author referred to is Burke (not Timothy).
- Caption of figure 6 (d): device type missing?

Reviewer #2 (Remarks to the Author):

The manuscript entitled “Correct Determination of CT State Energy from Luminescence Spectra in Organic Solar Cells by Means of Optical Out Coupling” presents the experimental measurement of EL spectra and theoretical analysis for the proper correction of the obtained spectra in order to more precisely determine the CT state energy.

After evaluating the revised manuscript and SI, I think this work should be published at Nat. Comm. for the reasons as follows.

1. Despite still less popular than absorption measurements, EL has become the mainstream characterization method in OPV research. In addition, CT state energy is a key parameter for understanding the working principle of OPV devices. Lots of analysis on device physics of the state-of-the-art OPV devices, such as the energy loss analysis of a high-VOC device, requires the precise determination of CT state energy. Therefore, it is critical to have such a paper to address the accuracy of using EL spectra to extract CT state energy.

2. This work is internally consistent. The conclusions were drawn based on thoughtful comparisons among systems with control samples provided. In the revised manuscript, the additional experiments the authors performed are crucial for the completeness of this work. Particularly, the authors added a high-performance non-fullerene system with an emission closer to its optical bandgap. They found that

the outcoupling factor effect is not negligible for this system and their methodology successfully addressed the shift of the EL spectra caused by layer thickness variation in a given device configuration. The comparison of the CT state energy values in the SI makes a clear demonstration about the necessity of their spectra correction (at least for the systems the authors studied). I suggest that this table be moved to the main text.

In short, the additional data in the revised manuscript provides their method with better generality and makes their conclusion more solid. I suggest that the authors shorten the theoretical background a little to make the introduction more concise, which is important for readers with different backgrounds to follow.

A comment to the authors (may not be necessary to address for this manuscript):

Measuring the sub-gap EQE, i.e. long wavelength region, has been used by multiple groups to probe the CT state energy. Optical microcavities and PAL thicknesses could also play an important role similar to EL experiments. Could similar analysis/corrections be carried out in the absorption based experiments?

REVIEWERS' COMMENTS:

Reviewer #1 (Remarks to the Author):

Authors have revised their paper substantially based on the referee comments. They have extended their study to real state of the art devices and they have clarified why it is important to include microcavity effects: Even though these effects are widely known in OLED research, they have been neglected for the extraction of CT state properties in OPV. Therefore, I feel the work is sufficiently important for publication in nature communications. In the revised version the impact on the determination of the CT state energy is now much more clear, although a bit hidden in the supporting information. It would be nice if the authors could put a number (0.1 eV but not 0.5 eV ?, and there is still a scattering of 50 to 100 meV on the CT state energy of corrected spectra) on the error made in the determination of the CT state energy when neglecting cavity effects, and mention these numbers in the main text. The errors seems to be smaller when thin films are used.

The table with the results from fitting was moved to the main text and some discussion on the numbers and the errors was added.

Minor:

- In the paragraphs on “CT state properties and photovoltage”, above equation (3), the last name of the author referred to is Burke (not Timothy).

Name was corrected.

- Caption of figure 6 (d): device type missing?

device type added

Reviewer #2 (Remarks to the Author):

The manuscript entitled “Correct Determination of CT State Energy from Luminescence Spectra in Organic Solar Cells by Means of Optical Out Coupling” presents the experimental measurement of EL spectra and theoretical analysis for the proper correction of the obtained spectra in order to more precisely determine the CT state energy.

After evaluating the revised manuscript and SI, I think this work should be published at Nat. Comm. for the reasons as follows.

1. Despite still less popular than absorption measurements, EL has become the mainstream characterization method in OPV research. In addition, CT state energy is a key parameter for

understanding the working principle of OPV devices. Lots of analysis on device physics of the state-of-the-art OPV devices, such as the energy loss analysis of a high-VOC device, requires the precise determination of CT state energy. Therefore, it is critical to have such a paper to address the accuracy of using EL spectra to extract CT state energy.

2. This work is internally consistent. The conclusions were drawn based on thoughtful comparisons among systems with control samples provided. In the revised manuscript, the additional experiments the authors performed are crucial for the completeness of this work. Particularly, the authors added a high-performance non-fullerene system with an emission closer to its optical bandgap. They found that the outcoupling factor effect is not negligible for this system and their methodology successfully addressed the shift of the EL spectra caused by layer thickness variation in a given device configuration. The comparison of the CT state energy values in the SI makes a clear demonstration about the necessity of their spectra correction (at least for the systems the authors studied). I suggest that this table be moved to the main text.

The table with the fit results was moved to the main text.

In short, the additional data in the revised manuscript provides their method with better generality and makes their conclusion more solid. I suggest that the authors shorten the theoretical background a little to make the introduction more concise, which is important for readers with different backgrounds to follow.

We have critically reviewed that section and found that if anything would be deleted the introduction would be incomplete. We know that this is always a compromise and please note that another referee demanded a more comprehensive description of the theoretical background when reviewing the previous version of the manuscript.

A comment to the authors (may not be necessary to address for this manuscript):

Measuring the sub-gap EQE, i.e. long wavelength region, has been used by multiple groups to probe the CT state energy. Optical microcavities and PAL thicknesses could also play an important role similar to EL experiments. Could similar analysis/corrections be carried out in the absorption based experiments?

Indeed this is expected to be necessary. Further, please note that we had amended the following sentence in the section 'Conclusion' (now renamed to 'Discussion'): It is important to note that the described correction procedure is required for the correct analysis of CT state absorption spectra.